# LATO: 3D Mesh Flow Matching with Structured TOpology Preserving LAtents

**Tianhao Zhao** [1][*]  **Youjia Zhang** [1][*]  **Hang Long** [1]  **Jinshen Zhang** [1]  **Wenbing Li** [1]  **Yang Yang** [1]
**Gongbo Zhang** [2]  **Jozef Hladký** [3]  **Matthias Nießner** [4]  **Wei Yang** [1][,][†]

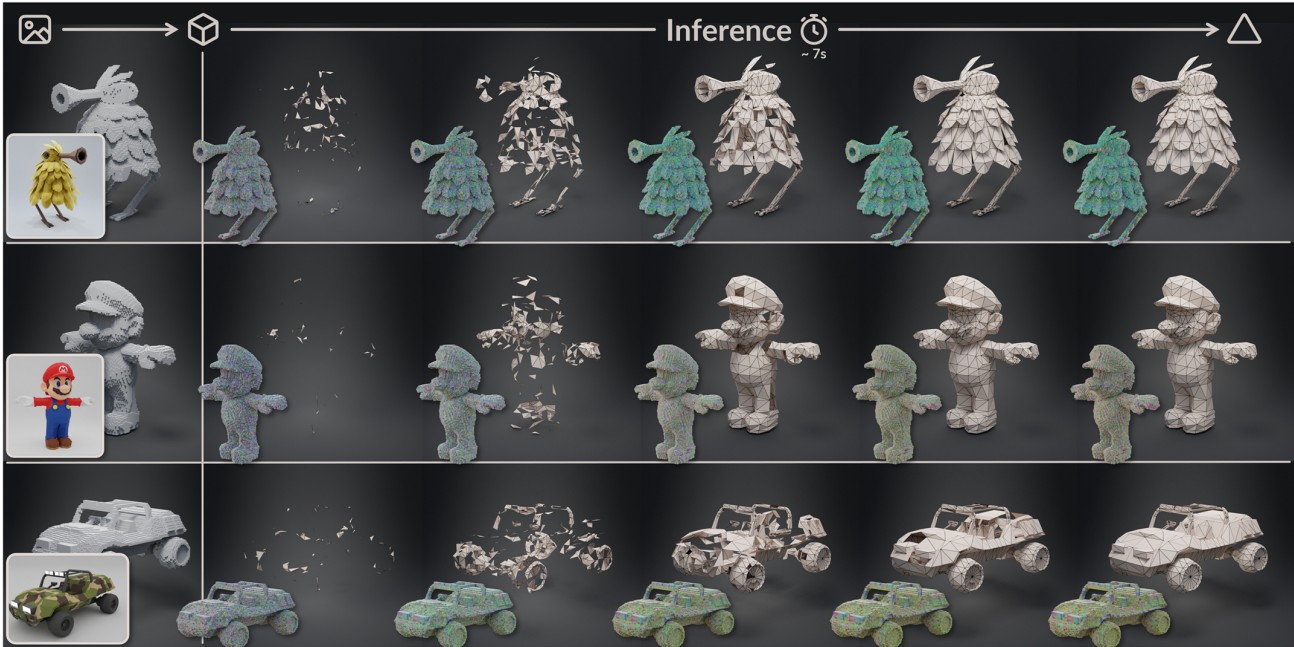

*Figure 1.* We present **LATO**, a topology-preserving sparse voxel representation for explicit 3D mesh generation. LATO first synthesizes a sparse voxel structure, followed by the progressive denoising of the latent **T-Voxels**. The mesh gradually materializes, as holes close and the complete topology is recovered.

## Abstract

In this paper, we introduce LATO, a novel topology-preserving latent representation that enables scalable, flow matching-based synthesis of explicit 3D meshes. LATO represents a mesh as a Vertex Displacement Field (VDF) anchored on surface, incorporating a sparse voxel Variational Autoencoder (VAE) to compress this explicit signal into a structured, topology-aware voxel latent. To decapsulate the mesh, the VAE decoder progressively subdivides and prunes latent voxels to instantiate precise vertex locations. In the end, a dedicated connection head queries the voxel latent to predict edge connectivity between vertex pairs directly, allowing mesh topology to be recovered without isosurface extraction or heuristic meshing. For generative modeling, LATO adopts a two-stage flow matching process, first synthesizing the structure voxels and subsequently refining the voxel-wise topology features. Compared to prior isosurface/triangle-based diffusion models and autoregressive generation approaches, LATO generates meshes with complex geometry, well-formed topology while being highly efficient in inference. Project page: https://tianhaozhao668.github.io/LATO/

[*]Equal contribution . [1]Huazhong University of Science and Technology [2]Peking University [3]Independent Researcher [4]Technical University of Munich. Correspondence to: Wei Yang <weiyangcs@hust.edu.cn>.

*Proceedings of the 43$^{rd}$ International Conference on Machine Learning*, Seoul, South Korea. PMLR 306, 2026. Copyright 2026 by the author(s).

## 1. Introduction

Recent advances in large-scale 3D generative modeling have paved the way for rapid content creation across domains

ranging from virtual reality to industrial design. A prevalent and successful paradigm relies on a two-stage process: a 3D Variational Autoencoder (VAE) first compresses shape into a compact latent space, followed by a latent diffusion model that models the distribution for generative sampling. Currently, the vast majority of these methods decode latents into implicit fields, such as Signed Distance Functions (SDFs) or occupancy fields, relying on isosurfacing algorithms like Marching Cubes (Lorensen & Cline, 1998) to extract the final surface mesh. A critical differentiator among state-of-the-art methods lies in the spatial inductive bias of their latents. One line of work represents shapes as vecset latents (e.g., 3DShape2VecSet (Zhang et al., 2023), CLAY (Zhang et al., 2024), Michelangelo (Zhao et al., 2023), TripoSG (Li et al., 2025a), and Hunyuan3D (Zhao et al., 2025b)), while another family of methods leverages sparse voxel based representations (e.g., TRELLIS (Xiang et al., 2025b;a), Direct3D-S2 (Wu et al., 2026), Sparc3D (Li et al., 2025b), LATTICE (Lai et al., 2025)) to enforce stronger spatial structure. Despite this implicit-diffusion paradigm excels at capturing overall geometry, both families share a critical limitation: they do not expose explicit mesh topology. Consequently, post-decoding meshing yields overly dense, irregular triangulations that deviate from artist-crafted topology, rendering them unsuitable for direct downstream rigging, deformation, and game-engine deployment. Furthermore, the reliance on implicit fields imposes a strict "watertightness" assumption on the training data, forcing the discarding of a massive fraction of open surfaces, non-manifold assets and exacerbating the data scarcity problem in 3D generation. Concurrently, TRELLIS.2 (Xiang et al., 2025a) utilizes a dual-grid representation that allows training on open surfaces and non-manifold geometries; however, its topology remains derived from grid-based rules rather than learned artist intent.

To address this topological degradation, recent methods attempt to model mesh connectivity explicitly. Diffusion-based approaches, including PolyDiff (Alliegro et al., 2023), MeshCraft (He et al., 2025a) and MeshFlow (Li et al., 2026), transform face-level or vertex-level features into continuous representations, while autoregressive models serialize the mesh into sequences to learn next-token distributions (Siddiqui et al., 2024; Chen et al.; 2025; 2024; Hao et al., 2024; Weng et al., 2025b; Tang et al., 2025; Song et al., 2026; Zhao et al., 2025a; Kim et al., 2025; Liu et al., 2026a; Lin et al., 2026; Xu et al., 2026). However, these explicit methods hit a computational ceiling: generating meshes with rich geometric detail inevitably yields excessively long token streams or high-dimensional feature spaces. To cope with memory constraints, these methods often resort to training on truncated sequences, resulting in fragmented components and broken surfaces.

To overcome this dichotomy between scalability and ex-

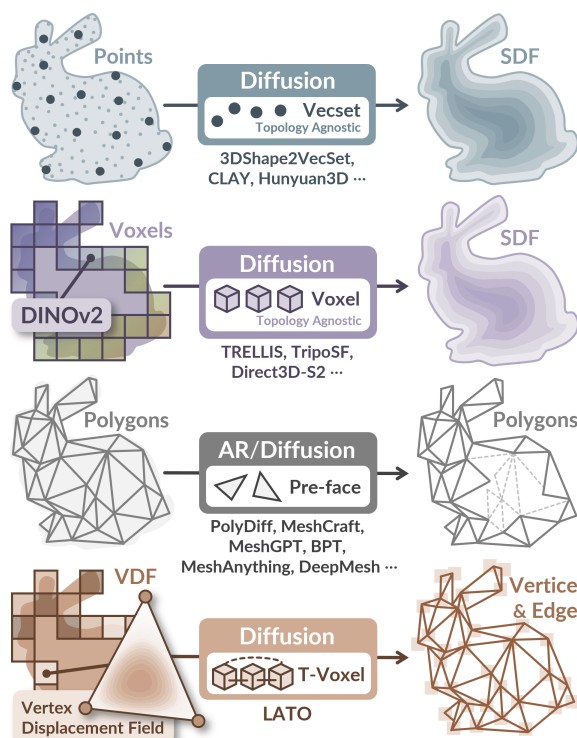

*Figure 2.* **LATO vs. Existing Paradigms.** Mainstream topology-agnostic approaches utilize vecset or voxel latents decoded into implicit fields (e.g., SDF), relying on Marching Cubes for mesh extraction. Conversely, explicit mesh generation methods adopt per-face latents via autoregressive or diffusion models, but suffer from severe memory bottlenecks. LATO proposes **T-Voxels** latents to explicitly model topology, enabling the direct generation of artist-friendly meshes.

plicit topological learning, we introduce **LATO**, a novel topology-preserving sparse voxel representation. LATO directly exposes explicit mesh connectivity to the generative modeling pipeline, unlocking the ability to train on open-surface and non-manifold assets. LATO represents the mesh as a Vertex Displacement Field (VDF) anchored to the surface geometry. Specifically, we sample points on the mesh and embed them with displacement vectors pointing to the vertices of the faces they occupy. By voxelizing and pooling these features, our VAE encodes these topology-infused structural features into a sparse voxel latent, which we term **T-Voxels**. Unlike vecset or sparse voxel latent, T-Voxels do not merely imply surface presence; they encode the spatial distribution and connectivity of mesh vertices, effectively serving as a structural prior for topology. During decoding, the VAE progressively subdivides latent voxels and utilizes a learnable pruning mechanism to instantiate precise vertex locations. Simultaneously, a dedicated connection head queries the T-Voxels to directly predict connectivity between vertex pairs, where we reasonably assume that triangular faces are closed 3-cycles within the graph. For generative modeling, LATO first synthesizes coarse structure voxels

and subsequently refining the topology features, inspired by TRELLIS (Xiang et al., 2025b;a). By unifying structural voxel with topology-preserving features, LATO sets a new paradigm for high-fidelity, scalable, and explicit 3D mesh generation. Extensive experiments demonstrate that LATO generates well-formed topologies comparable to explicit autoregressive methods, while maintaining the high geometric fidelity and inference efficiency of implicit models.

## 2. Related Work

**3D Shape Representations.** The efficacy of 3D generation hinges on the underlying data representation. Early approaches utilized point clouds processed via per-point MLPs (Qi et al., 2017a;b; Wang et al., 2019; Guo et al., 2021) or voxel grids (Brock et al., 2016; Dai et al., 2017; Girdhar et al., 2016; Wu et al., 2016; 2015; Choy et al., 2016). To mitigate memory complexity, octree-based sparsification was introduced to focus computation on active regions (Häne et al., 2017; Tatarchenko et al., 2017; Wang et al., 2017; 2018). More recently, implicit neural fields have become the dominant paradigm, representing shapes as continuous functions mapping coordinates to scalars (e.g., occupancy (Mescheder et al., 2019; Chen & Zhang, 2019), SDF (Jiang et al., 2020; Michalkiewicz et al., 2019)) or vectors (e.g., radiance fields (Chan et al., 2022)).

**3D Diffusion Models.** Building upon shape representations, 3D diffusion models generally fall into two categories: 2D-lifting and 3D-native generation. The 2D-lifting paradigm leverages pre-trained 2D diffusion priors via Score Distillation Sampling (SDS) (Poole et al., 2023; Lin et al., 2023; Wang et al., 2023) or multi-view reconstruction (Liu et al., 2023; Shi et al., 2023; Long et al., 2024; Liu et al., 2024). LRM (Hong et al., 2024; Tang et al., 2024; Xu et al., 2024) further accelerate this by regressing NeRFs or 3D Gaussians (Kerbl et al., 2023) directly from images in a feedforward manner. In contrast, 3D-native diffusion learns generative distributions directly in a compressed 3D latent space. 3DShape2VecSet (Zhang et al., 2023) and CLAY (Zhang et al., 2024) encode shapes into vecset latent, while recent works like TRELLIS (Xiang et al., 2025b;a), XCube (Ren et al., 2024), TripoSF (He et al., 2025b), Direct3D-S2 (Wu et al., 2026), and Sparc3D (Li et al., 2025b) utilize structured sparse voxel latents to improve geometric fidelity. LATTICE (Lai et al., 2025) further hybridizes these by localize vecset to surface voxels. Crucially, these approaches focus on surface recovery, deriving mesh connectivity implicitly via isosurfacing algorithms rather than explicitly modeling it. This results in dense, irregular triangulations. Concurrently, TRELLIS.2 (Xiang et al., 2025a) utilizes a dual-grid representation to improve meshing robustness; however, its topology remains derived from grid-based rules rather than learned artist intent. In contrast, LATO prioritizes the gener-

ation of explicit, artist-intended topology directly from the latent space.

**Topology-preserving Mesh Generation.** To address the topology vanishment, a separate line of work treats mesh generation as a sequence modeling problem. PolyGen (Nash et al., 2020) and MeshGPT (Siddiqui et al., 2024) tokenize mesh faces and utilize autoregressive transformers to predict the next token. Subsequent research has focused on optimizing this tokenization to handle higher resolutions: MeshAnythingV2 (Chen et al., 2025) introduces Adjacent Mesh Tokenization, EdgeRunner (Tang et al., 2025) and Mesh Silksong (Song et al., 2026) leverage graph-traversal ordering, and BPT (Weng et al., 2025b) adopts a patch-wise strategy. Others, like MeshTron (Hao et al., 2024), MeshXL (Chen et al., 2024), operate on vertex coordinates directly. DeepMesh (Zhao et al., 2025a) and MeshRFT (Liu et al., 2026b) introduce reinforced fine-tuning to improve topology quality. In contrast, PolyDiff (Alliegro et al., 2023) and MeshCraft (He et al., 2025a) use per-face latents, and leverage diffusion model to generate the faces directly. Despite these advances, these methods face a fundamental bottleneck: the representation scales quadratically with mesh complexity. To satisfy memory constraints, training is often performed on truncated sequences, which inevitably leads to broken surfaces and fragmented components.

## 3. Method

We propose LATO, a topology-preserving mesh generation framework as shown in Fig. 3. LATO utilizes a VDF representation, where sampled surface points encode displacement vectors to their constituent vertices (Sec. 3.1). These features are compressed into a topology preserving sparse voxel latent, termed **T-Voxels**. To reconstruct the mesh, the VAE decoder employs hierarchical voxel subdivision with learnable vertex pruning to instantiate precise locations, alongside a parallel connection head that directly predicts edge connectivity (Sec. 3.2). For generation, we adopt a cascaded diffusion strategy, following TRELLIS (Xiang et al., 2025b), that first synthesizes coarse geometry voxels followed by fine-grained topological features (Sec. 3.3).

### 3.1. Vertex Displacement Field

Prevailing 3D diffusion pipelines utilize latents derived from point clouds or occupancy voxels, which are fundamentally topology-agnostic: i.e., they encode the continuous surface boundary but discard the discrete vertex distribution and mutual connectivity that constitute the explicit mesh topology. To bridge this gap, we propose the Vertex Displacement Field (VDF), denoted as $\mathcal{F}$, a representation designed to preserve both geometry and topology. Let a mesh be defined as $\mathcal{M} = (\mathbf{V}, \mathbf{F})$, where $\mathbf{V} = \{\mathbf{v}_i, i = 1 \ldots \mathrm{N}\} \subset \mathbb{R}^3$ represents the set of vertices, and $\mathbf{F} = \{\mathbf{f}_j, j = 1 \ldots \mathrm{M}\}$ repre-

## 3D Mesh Encoding & Decoding

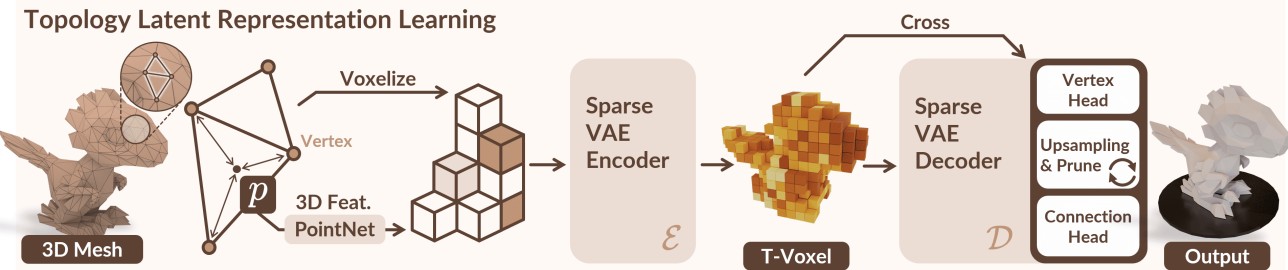

## 3D Mesh Generation

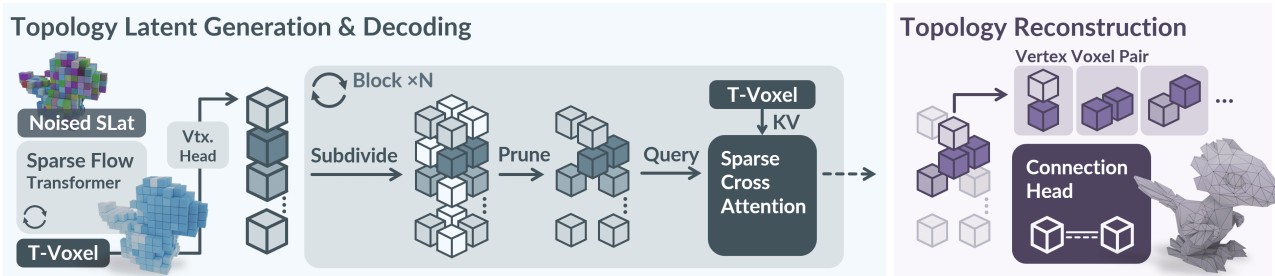

*Figure 3.* **Overview of the LATO pipeline.** We explicitly encode mesh topology by sampling surface points infused with relative displacement to their enclosing face vertices (Vertex Displacement Field, VDF). These dense features are aggregated and compressed via a sparse voxel VAE into a structured latent representation, termed **T-Voxels**. To reconstruct the mesh, the T-Voxels undergo hierarchical subdivision and learnable pruning to precisely instantiate high-resolution vertex locations. Simultaneously, a connection head predicts edge existence between vertex pairs, directly recovering the explicit mesh topology.

sents the set of triangular faces. Each face $\mathbf{f} = [f_0, f_1, f_2]$ is a triplet of indices referencing vertices in $\mathbf{V}$. For any point $\mathbf{p}$ sampled on the surface of face $\mathbf{f}$, we define the field value $\mathcal{F}(\mathbf{p})$ as the set of relative displacement vectors pointing to the constituent vertices of face $\mathbf{f}$:

$$\mathcal{F}(\mathbf{p}) = \Big\{ \mathbf{v} - \mathbf{p} \mid \mathbf{v} \in \{\mathbf{v}_{f_0}, \mathbf{v}_{f_1}, \mathbf{v}_{f_2}\} \Big\}, \mathbf{p} \in \mathbf{f}. \quad (1)$$

Unlike points or occupancy voxels, $\mathcal{F}$ explicitly encodes the local structural layout of the triangulation: the zero-magnitude displacements ($\|\mathbf{v} - \mathbf{p}\| \to 0$) precisely localize the vertex positions, while the discontinuities in $\mathcal{F}$ composition delineate the edges between vertex pairs.

**Compare with Naive Alternative.** While other formulations exist, they lack dense supervisory signal as VDF. A naive alternative is to directly embed surface points with semantic labels, e.g., classifying points as vertex, edge, or face. However, this approach suffers from two critical challenges: 1, vertices and edges are zero-measure sets relative to the continuous surface area, random sampling leads to extreme class imbalance resulting in poor supervision signals; 2, imposing discrete, categorical labels introduces abrupt discontinuities in the latent space, impeding the smooth gradient flow required for effective diffusion learning. In contrast, $\mathcal{F}$ provides a dense, continuous signal: every point on the face carries the explicit coordinate information required to reconstruct the enclosing vertices and their connectivity. We provide an empirical comparison in Fig. 9.

### 3.2. Sparse Voxel VAE

To compress VDF into a tractable representation for generative modeling, we employ a sparse voxel VAE to learn a probabilistic mapping between the explicit surface displacement features and a compact, structured latent space, denoted as **T-Voxels.** The architecture leverages a sparse convolutional/transformer backbone adapted from TREL-LIS (Xiang et al., 2025b) and TripoSF (He et al., 2025b) as encoder, but introduces a specialized decoding protocol designed to preserve topological fidelity.

Given a mesh $\mathcal{M}$, we uniformly sample a point cloud $\mathcal{P} = \{\mathbf{p}_k\}_{k=1}^K$ from the surface. For each point $\mathbf{p}_k$, we construct a topology-infused feature vector $\mathbf{x}_k = [\mathbf{p}_k, \mathcal{F}(\mathbf{p}_k), \mathbf{n}(\mathbf{p}_k)]$, where $\mathcal{F}(\mathbf{p}_k)$ represents the relative displacement vectors to the vertices, and $\mathbf{n}(\mathbf{p}_k)$ denotes the surface normal. The VAE is defined as a mapping from this feature set $\mathbf{X} = \{\mathbf{x}_k\}$ to a sparse latent grid $\mathbf{z}$ (T-Voxels), and subsequently to a reconstructed graph $\{\hat{\mathbf{v}}, \hat{\mathbf{e}}_{ij}\}$, where $\hat{\mathbf{v}}$ is the vertex location and $\hat{\mathbf{e}}_{ij}$ represents the connectivity between $\hat{\mathbf{v}}_i$ and $\hat{\mathbf{v}}_j$ (0 for no edge, 1 for edge existence):

$$\mathbf{z} = \mathcal{E}(\mathbf{X}), \quad \{\hat{\mathbf{v}}, \hat{\mathbf{e}}_{ij}\} = \mathcal{D}(\mathbf{z}), \quad (2)$$

where $\mathcal{E}$ and $\mathcal{D}$ denote the probabilistic encoder and decoder.

**Encoder.** To process the unstructured point features $\mathbf{X}$, we first spatially discretize $\mathbf{X}$ into a sparse voxel grid. To preserve local geometric details, we employ a Point-

Net layer (Qi et al., 2017a) within each occupied voxel $\mathbf{v}$. Specifically, for all points falling within voxel, we apply a shared MLP followed by a symmetric mean-pooling operation. These local voxel embeddings are then processed by a sparse transformer backbone to capture global shape context. Finally, the encoder predicts the mean and log-variance for each active voxel, from which the T-Voxel latent $\mathbf{z}$ is sampled via reparameterization.

The decoder takes the latent $\mathbf{z}$ as input to reconstruct the explicit vertex set $\{\hat{\mathbf{v}}\}$ and edge connection set $\{\hat{\mathbf{e}}_{ij}\}$.

**Vertex Set Prediction.** Directly regressing vertex coordinates is challenging due to the varying cardinality of mesh vertices across different shapes. To resolve this, we adopt a hierarchical subdivide and prune strategy that progressively increases the discretization resolution to locate vertices precisely. Specifically, given the coarse T-Voxels $\mathbf{z}$, we first apply a sparse transformer layer followed by an initial **Pruning Head**, a linear classifier that predicts an occupancy probability (indicates vertex presence) for each voxel. Voxels with probability $< 0.5$ are discarded to ensure sparsity. From this initialization, we cascade $L = 3$ stages of **Upsample-and-Refine** modules. Each module begins with a voxel subdivision layer, where every active parent voxel is subdivided into $2^3 = 8$ octants. To initialize the features of these new sub-voxels, we replicate the parent feature and apply a relative positional embedding followed a 3D convolution layer to disambiguate the local spatial semantics within the larger voxel. This is followed by a pruning head to remove empty sub-voxels, a sparse self-attention module to aggregate local context, and finally a cross-attention layer where the current sub-voxels serve as queries and the original global T-Voxels $\mathbf{z}$ serve as keys/values. The centroids of the voxels remaining after the final refinement stage constitute the predicted vertex set $\{\hat{\mathbf{v}}\}$.

**Connection Prediction Head.** To fully recover the topology, we need to predict the edge connection between each vertex pairs. For effective prediction, we first cross attend the vertex voxel features before final output layer with T-Voxels $\mathbf{z}$ to aggregate the vertex with global context feature. Then we concatenate the features of two vertices $\hat{\mathbf{v}}_i$ and $\hat{\mathbf{v}}_j$, and use a MLP to predict the connectivity, as:

$$\mathbf{h}_{\hat{\mathbf{v}}} = \text{CrossAttn}(\mathbf{h}_{\hat{\mathbf{v}}}, \mathbf{z}), \quad \hat{\mathbf{e}}_{ij} = \text{MLP}(\mathbf{h}_{\hat{\mathbf{v}}_i} \oplus \mathbf{h}_{\hat{\mathbf{v}}_j}). \quad (3)$$

A naive enumeration of all vertex pairs during training results in a quadratic complexity, which is computationally prohibitive. To alleviate this issue, we adopt a sampling-based training strategy. Specifically, all vertex pairs corresponding to ground-truth edges are treated as positive samples and will be included in the training sample. For each vertex, we further sample its $N_n$ neighboring vertices and randomly select $N_r$ vertices from the entire vertex set to construct candidate pairs. Furthermore, since edges are undirected, we enforce permutation invariance in the connection

head. For each vertex pair $\hat{\mathbf{v}}_i$ and $\hat{\mathbf{v}}_j$, we concatenate the two vertices in both orders, i.e., $\mathbf{h}_{\hat{\mathbf{v}}_i} \oplus \mathbf{h}_{\hat{\mathbf{v}}_j}$ and $\mathbf{h}_{\hat{\mathbf{v}}_j} \oplus \mathbf{h}_{\hat{\mathbf{v}}_i}$, and pass them through the connection head independently. The final prediction is obtained by averaging the two outputs, ensuring that the edge prediction is invariant to the ordering of vertices. During inference, we exhaustively evaluate all vertex pairs to ensure that the model can correctly identify all existing edges.

**Loss Function.** We train the VAE in an end-to-end manner, optimizing a composite objective that enforces both geometric fidelity and topological precision.

To ensure the decoder correctly identifies active regions across scales, we supervise the occupancy probability at every hierarchical level $l$. The pruning loss is defined as:

$$\mathcal{L}_{\text{prune}} = \sum_{l=1}^{L-1} \mathbb{E}_{\mathbf{v} \in \mathcal{M}} \big[ \text{BCE}(\hat{o}(\mathbf{v}, l), o(\mathbf{v}, l)) \big], \quad (4)$$

where $o(\mathbf{v}, l) \in \{0, 1\}$ denotes the ground truth occupancy of voxel $\mathbf{v}$ at level $l$, and $\hat{o}$ is the predicted probability.

At the finest level, identifying the exact voxels that contain vertices is a highly imbalanced classification task (as vertices occupy a negligible fraction of the total volume). To address this, we employ the Asymmetric Loss (Ridnik et al., 2021) for the final vertex prediction at level $L$:

$$\mathcal{L}_{\text{vtx}} = \mathbb{E}_{\mathbf{v} \in \mathcal{M}} \big[ o(\mathbf{v}, L) A^+ + (1 - o(\mathbf{v}, L)) A^- \big], \quad (5)$$

where $A^+ = [1 - \hat{o}(\mathbf{v}, L)]^{\gamma_+} \cdot \log(\hat{o}(\mathbf{v}, L))$ and $A^- = [\hat{o}(\mathbf{v}, L)]^{\gamma_-} \cdot \log(1 - \hat{o}(\mathbf{v}, L))$ define the loss for positive samples and negative samples respectively. By setting $\gamma_- > \gamma_+$ ($\gamma_-$ is 4 and $\gamma_+$ is 0, in our implementation), we downweight the negative samples to focus learning on the positive vertex locations. To recover the topology, we explicitly supervise the connection head. For every pair of candidate vertices $\{\mathbf{v}_i, \mathbf{v}_j\}$, we minimize the error in edge prediction:

$$\mathcal{L}_{\text{conn}} = \mathbb{E}_{\{\mathbf{v}_i, \mathbf{v}_j\} \sim \mathcal{M}} \big[ \text{BCE}(\hat{e}_{ij}, e_{ij}) \big], \quad (6)$$

where $e_{ij} = 1$ if an edge exists between $\mathbf{v}_i$ and $\mathbf{v}_j$, and 0 otherwise. The final objective is a weighted summation of the reconstruction terms and the KL regularization for the variational latent space:

$$\mathcal{L}_{\text{total}} = \mathcal{L}_{\text{prune}} + \mathcal{L}_{\text{vtx}} + \mathcal{L}_{\text{conn}} + \beta \mathcal{L}_{\text{KL}}. \quad (7)$$

### 3.3. Explicit Mesh Flow Matching

Building upon our topology-preserving VAE, we implement a flow matching pipeline for image to explicit mesh generation. Following TRELLIS (Xiang et al., 2025b), we factorize the generative process into two sequential stages: geometric structure synthesis and topological feature generation. The first component predicts the spatial occupancy

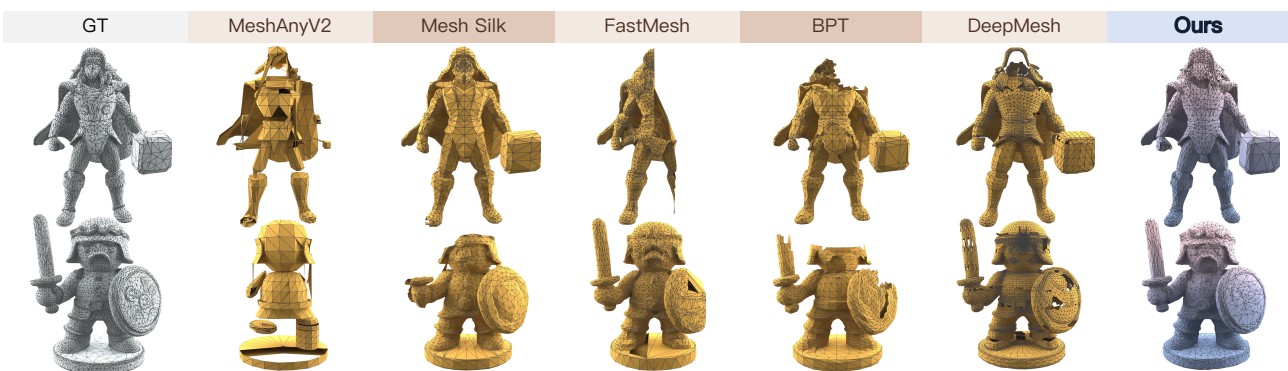

*Figure 4.* **Geometry conditioned generation comparison**. We compare our method against state-of-the-art baselines. Existing methods result in either incomplete reconstructions or excessively dense and irregular topology. In contrast, our LATO generates hole-free meshes with well-formed topology suitable for downstream applications.

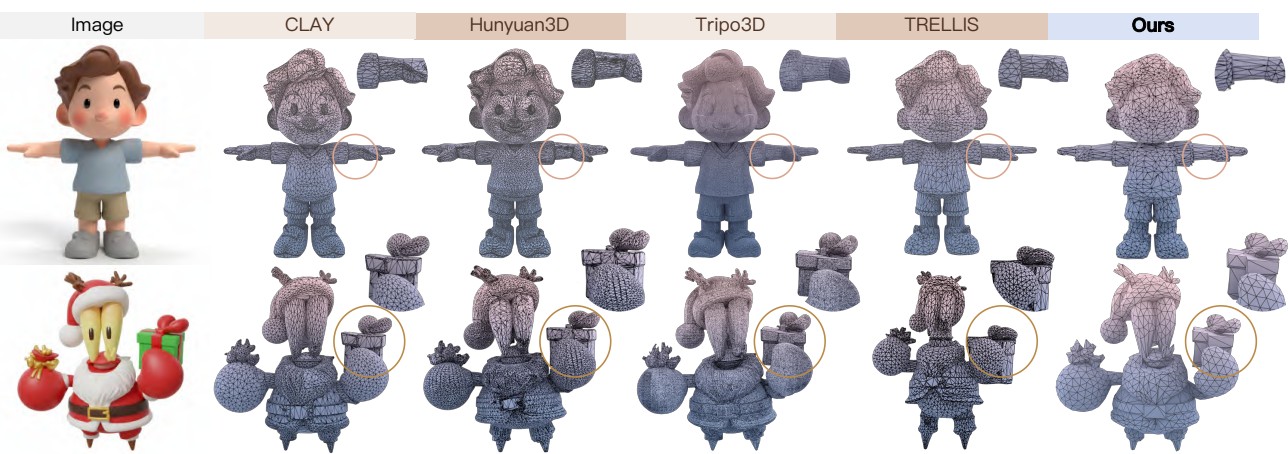

*Figure 5.* **Topology comparison with implicit foundation models**. We qualitatively compare against industrial-scale implicit baselines. While these foundation models possess scale and training resource advantages, their reliance on implicit field yields dense, irregular triangulation. In contrast, LATO generates artist-friendly edge flows.

*Table 1.* **VAE reconstruction comparison.** We compare reconstruction quality on dense/artistic meshes. Values are reported as dense/artistic for each metric; "−" denotes OOM.

| Method | CD(L2)↓ | CD(L1)↓ | HD↓ | |NC| ↑ |
|---|---|---|---|---|
| MeshGPT | 0.042/0.050 | 0.061/0.069 | 0.070/0.096 | 0.824/0.831 |
| PivotMesh | −/0.129 | −/0.175 | −/0.263 | −/0.614 |
| MeshCraft | 0.150/0.243 | 0.212/0.327 | 0.282/0.299 | 0.531/0.625 |
| **Ours** | **0.042/0.040** | **0.061/0.056** | **0.065/0.092** | **0.847/0.834** |

distribution from an image. We modify and fine-tune the pre-trained TRELLIS structure model with binary occupancy supervision to generate a sparse voxel occupancy at $128^3$ resolution. Our fine-tuning strategy explicitly incorporates open-surface and non-manifold assets, enabling the model to learn the diverse geometric distributions supported by our LATO representation. Once the geometric structure is established, the second stage serves as a topology feature generator. This model takes the sparse occupied voxels from

the first stage as spatial anchors and synthesizes the T-Voxel features. Formally, we train a flow matching transformer conditioned on vertex number $\mathbf{c}_v = \log(\mathrm{N}_v)$.

$$\mathcal{L}_{\text{TFlow}}(\theta) = \mathbb{E}_{\mathbf{z}_0, t, \boldsymbol{\epsilon}} \left( \left\| v_\theta(\mathbf{z}_t, t, \mathbf{c}_v) - (\boldsymbol{\epsilon} - \mathbf{z}_0) \right\|_2^2 \right). \quad (8)$$

During inference, we first sample the structure flow model to determine the active voxels, then generate the topology features for each voxel. Finally, the populated T-Voxels is passed to decoder $\mathcal{D}$, which directly instantiates vertices and predicts connectivity to yield the final explicit mesh.

## 4. Experiments

### 4.1. Implementation Details

We sample 819,200 points per-mesh. The encoder utilizes a simplified PointNet with feature dimensions 1024, followed by mean pooling to aggregate features into a sparse voxel grid. This is processed by a Sparse Transformer (dimen-

| Image | Image to 3D |
|---|---|

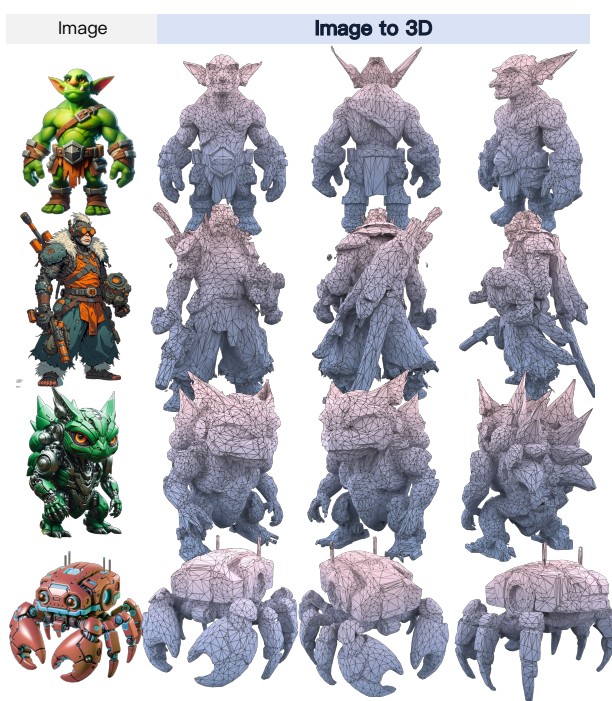

*Figure 6.* **Image to 3D generation results.**

*Table 2.* Quantitative comparison with topology-preserving generation methods on dense/artistic meshes (reported as dense/artistic).

| Method | CD(L2)↓ | HD↓ | \|NC\| ↑ |
|---|---|---|---|
| MeshAnythingv2 | 0.108/0.066 | 0.227/0.137 | 0.696/0.766 |
| MeshSilkSong | 0.062/0.052 | 0.145/0.101 | 0.784/0.818 |
| FastMesh | 0.064/0.048 | 0.110/0.102 | 0.757/0.813 |
| BPT | 0.059/0.052 | 0.107/0.088 | 0.811/0.824 |
| DeepMesh | 0.051/0.046 | 0.092/0.089 | 0.828/0.827 |
| **Ours** | **0.043/0.044** | **0.084/0.081** | **0.832/0.835** |

sion 512, 8 heads) to produce the **T-Voxel** latent at $128^3$ resolution with 16 channels. The decoder mirrors this structure with a bottleneck dimension of 512. The structure flow model is adapted from TRELLIS (Xiang et al., 2025b) and TripoSF (He et al., 2025b), generates structure voxels from a $32^3$ latent. The topology latent model is a conditional Flow Matching Transformer that regresses the continuous 16-channel T-Voxel features. We curate a dataset of approximately 400K assets from TRELLIS-500K (Xiang et al., 2025b), Objaverse (Deitke et al., 2023), Toys4K (Stojanov et al., 2021), and ABO (Collins et al., 2022), retaining non-watertight meshes to ensure topological diversity. We use the AdamW (Loshchilov & Hutter) optimizer with a cosine learning rate schedule decaying from $10^{-4}$ to $10^{-5}$. The VAE is trained on 8 NVIDIA H100 GPUs for 4 days with a per-GPU batch size of 8. Subsequently, the two flow matching models are trained on frozen latents for 7 days in total

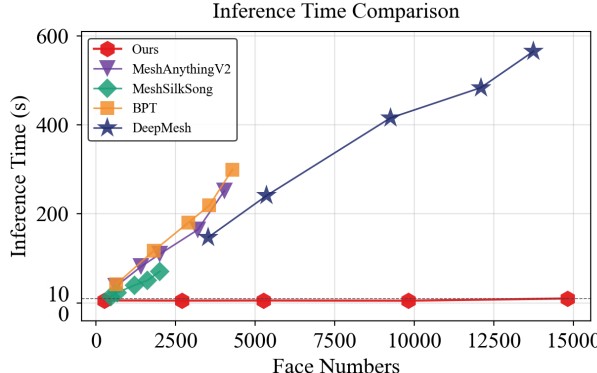

*Figure 7.* **Inference time comparison.** The inference time evaluation is conducted on a single H100 GPU. LATO maintains rapid generation (3∼10s), whereas autoregressive methods exhibit prohibitive temporal scaling, often requiring minutes for high-fidelity outputs.

*Table 3.* Ablation study on VAE input. We evaluated the effects of various VAE inputs on reconstruction performance.

| Method | CD(L2)↓ | CD(L1)↓ | HD↓ | \|NC\| ↑ |
|---|---|---|---|---|
| w/o $\mathbf{n}(\mathbf{p}_k)$ | 0.040 | 0.057 | 0.076 | 0.832 |
| w/o $\mathcal{F}(\mathbf{p})$ | 0.043 | 0.061 | 0.092 | 0.822 |
| $K = 81,920$ | 0.040 | 0.056 | 0.078 | 0.832 |
| w/ $\mathrm{norm}(\mathcal{F}(\mathbf{p}))$ | 0.042 | 0.059 | 0.082 | 0.827 |
| w/ $\mathrm{norm}_\tau(\mathcal{F}(\mathbf{p}))$ | 0.041 | 0.058 | 0.081 | 0.832 |
| **Ours(Full model)** | **0.039** | **0.056** | **0.075** | **0.837** |

with an effective batch size of 128 via gradient accumulation. We evaluate on two sets: artist mesh, comprising 200 hold-out assets from G-Objaverse, Toys4K, and ShapeNet; and dense mesh, consisting of 200 high-fidelity samples generated by TRELLIS (Xiang et al., 2025b).

### 4.2. Quantitative Analysis

**VAE Reconstruction.** We compare our continuous voxel VAE against discrete, per-face explicit baselines (MeshGPT (Siddiqui et al., 2024), PivotMesh (Weng et al., 2025a), MeshCraft (He et al., 2025a)). We measure Chamfer Distance (CD), Hausdorff Distance (HD), and Normal Consistency (NC). As shown in Tab. 1, LATO outperforms these methods across all metrics. **Geometry-Conditioned Generation.** We benchmark LATO against topology-aware autoregressive methods, including MeshAnything-v2 (Chen et al., 2025), BPT (Weng et al., 2025b), FastMesh (Kim et al., 2025), Mesh Silksong (Song et al., 2026), and DeepMesh (Zhao et al., 2025a). For fairness, we derive our voxel condition from the same source geometry used for baseline point clouds. As summarized in Tab. 2, LATO achieves state-of-the-art alignment scores on both dense and artistic mesh benchmarks, validating the robust representa-

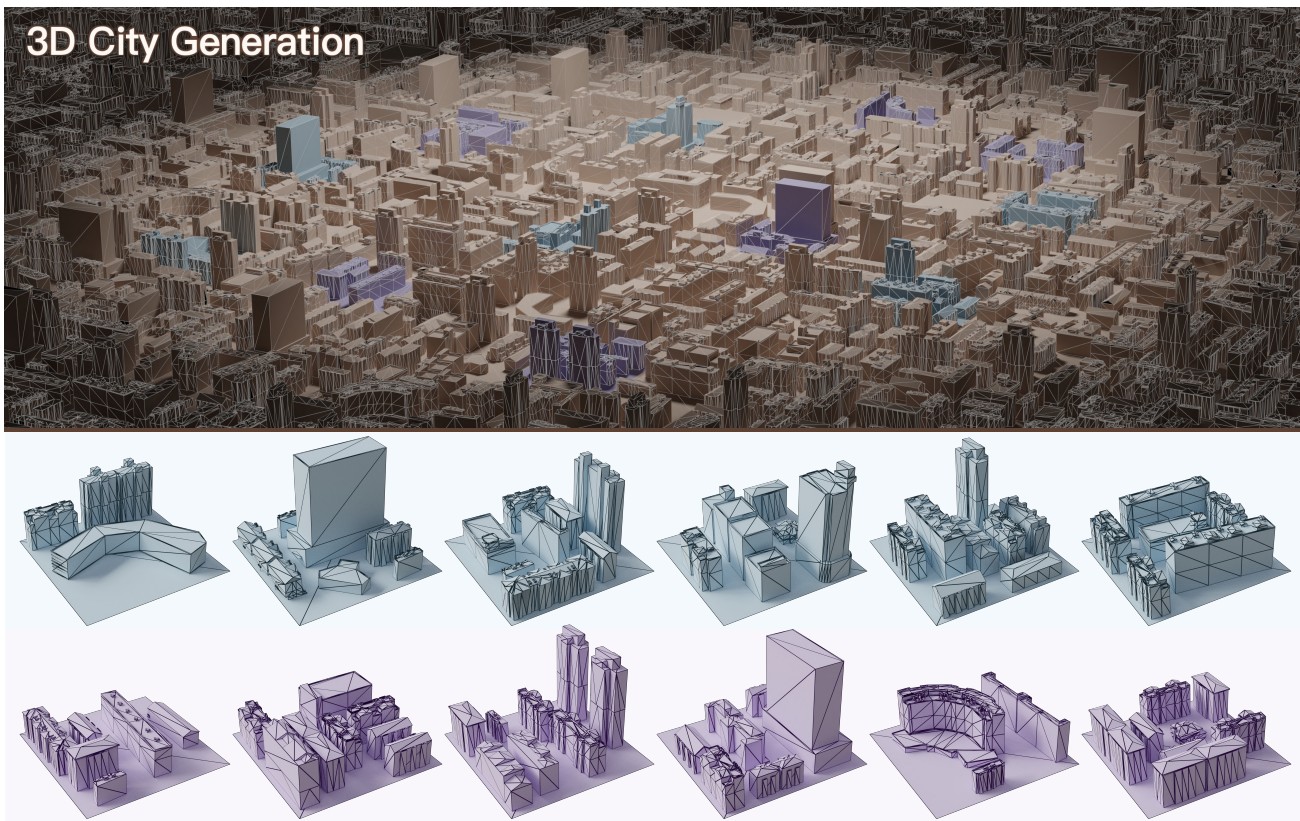

*Figure 8.* **Scene-scale urban generation.** We demonstrate LATO's extensibility on 3D city data collected from website, we first synthesizes block-wise building and subsequently populates T-Voxel features. Because our sparse voxel is inherently compositional, these individual units can be seamlessly assembled into expansive, high-fidelity cityscapes.

tion power of T-Voxels.

**Topology Quality.** Following the evaluation protocol in Mesh-RFT (Liu et al., 2026b), we assess topology quality using the **Topology Score**, which converts meshes into quadrilateral representations and measures how well-proportioned the resulting quad faces are. We compare the topology quality of our method against both autoregressive and diffusion-based models, as shown in Tab. 4. LATO achieves the highest topology score among all compared methods.

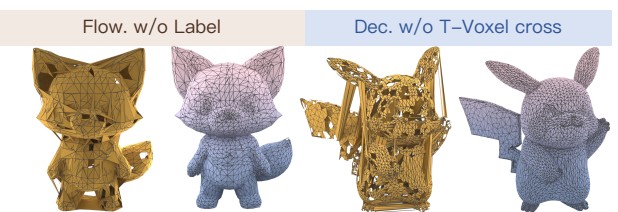

*Figure 9.* **Ablation study on VDF and T-Voxel cross-attention in decoder.** Left: we compare the voxel label (vertex, edge, face) input with our VDF input. The discrete label hinges the continuous learning using flow matching model. Right: Remove the T-Voxel cross-attention in decoder introduces hole and mismatches.

**Inference Efficiency.** Unlike autoregressive baselines constrained by sequential prediction, LATO employs a parallel flow matching solver that decouples latency from mesh complexity. As shown in Fig. 7, LATO maintains rapid generation (3∼10s), whereas autoregressive methods exhibit prohibitive temporal scaling, often requiring minutes for high-fidelity outputs.

### 4.3. Qualitative Analysis

We compare qualitative results of LATO against topology-aware autoregressive methods, including MeshAnything-v2 (Chen et al., 2025), BPT (Weng et al., 2025b), FastMesh (Kim et al., 2025), Mesh Silksong (Song et al., 2026), and DeepMesh (Zhao et al., 2025a). The results are shown in Fig. 4, the autoregressive baselines generally produce holes due to truncated training strategy. **Topology Comparison vs. Implicit Models.** We qualitatively compare against industrial-scale implicit baselines (TREL-LIS (Xiang et al., 2025b), CLAY (Zhang et al., 2024), Hunyuan3D (Zhao et al., 2025b)). While these foundation models possess scale and training resource advantages, their reliance on implicit field yields dense, irregular triangula-

*Table 4.* Comparison of topology quality across different generation paradigms measured by Topology Score.

| Paradigm | Method | Topology Score↑ |
|---|---|---|
| Implicit | TRELLIS | 0.5264 |
| | Hunyuan3D | 0.2946 |
| | Tripo3D | 0.5146 |
| | CLAY | 0.5100 |
| AR | DeepMesh | 0.5500 |
| | FastMesh | 0.5470 |
| | BPT | 0.5415 |
| | MeshSilkSong | 0.5223 |
| T-Voxel | **LATO** | **0.5551** |

tion. In contrast, LATO generates artist-friendly edge flows (Fig. 5). **Image to 3D Mesh.** For image conditioned generation, we use our structure flow matching model to synthesize a structure voxels from images, and then use the topology latent flow matching model to generate the T-Voxel features. The generated results are shown in Fig. 6. **Vertex Number Conditioned Generation.** Fig. 10 shows the generated results with varying vertex number conditions $\mathbf{c}_v$, it clearly shows the face number tunes as we tone vertex number condition.

### 4.4. Ablation Study

**VAE input.** We compared the impact of different VAE inputs on reconstruction quality and found that our VDF achieves the best reconstruction results (Tab. 3). "w/o $\mathbf{n}(\mathbf{p}_k)$" removes normal vectors from the input. "w/o $\mathcal{F}(\mathbf{p})$" removes the displacement used to represent local geometric offsets from the vertices. Additionally, we analyze the effect of the number of sampled surface points ($K$). As expected, a larger $K$ leads to better reconstruction quality. Results with a smaller sampling number ($K = 81,920$) are reported in Tab. 3. In all of our experiments, $K$ is set to $819,200$. "w/ $\mathrm{norm}(\mathcal{F}(\mathbf{p}))$" applies normalization to each displacement vector in the field $\mathcal{F}(\mathbf{p})$. "w/ $\mathrm{norm}_\tau(\mathcal{F}(\mathbf{p}))$" applies normalization to a displacement vector only if its magnitude is larger than the threshold $\tau$. The last row corresponds to our full model using all input features.

**On VDF vs. labeled voxels.** We compared using explicit vertex-edge voxel versus using VDF as VAE inputs. We found that explicit edge voxels result in broken structures (Fig. 9 left).

**On connectivity inference strategy.** We compared a deterministic geometric algorithm with our learnable Connection Head. While the geometric algorithm suffices for reconstruction, it is brittle against the distribution gap present in generated samples.

**Application: City Synthesis.** To demonstrate LATO's ex-

tensibility, we apply it to architectural generation. Trained on the website collected urban dataset, our pipeline first synthesizes block-wise building envelopes via the structure model and subsequently populates them with detailed T-Voxel features. As our sparse voxel representation is inherently compositional, these individual units can be seamlessly assembled into expansive, high-fidelity cityscapes.

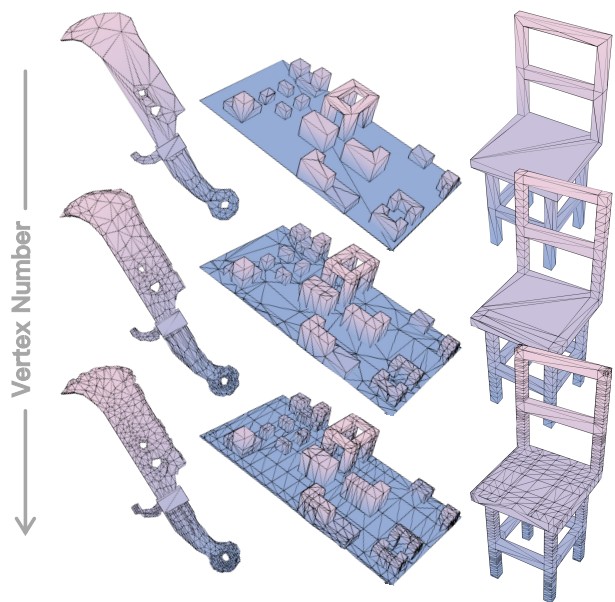

*Figure 10.* **Effect of vertex number condition.** As we increase the vertex number condition scaler $\mathbf{c}_v$, the model generates more vertices and triangles.

### 5. Conclusion

In this paper, we introduce LATO, a topology-preserving latent representation for scalable 3D mesh generation. Unlike prior implicit-field based approaches that discard explicit mesh connectivity, LATO jointly encodes geometric structure and mesh topology into a unified sparse voxel latent space. Extensive experiments demonstrate that LATO achieves strong reconstruction fidelity while producing superior topology quality compared with existing diffusion-based and autoregressive mesh generation methods. We believe LATO provides a promising direction toward scalable and topology-aware 3D asset generation.

**Limitations:** Despite various advantages, the VDF resolution remains constrained by the underlying sparse voxel grid, limiting LATO's ability to represent extremely small triangles or ultra-fine geometric details. In future work, we plan to incorporate octree-based representations to address resolution constraints to further enhance the generative capabilities and structural precision of the LATO framework.

## Acknowledgements

This research was supported by the National Natural Science Foundation of China (No. 62272184). The computational work was performed on the high-performance computing platform at Huazhong University of Science and Technology.

## Impact Statement

This paper presents work whose goal is to advance the field of Machine Learning. There are many potential societal consequences of our work, none which we feel must be specifically highlighted here.

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

## A. Pipeline Details

We sample point clouds from the mesh surface, including positions, normals, and Vertex Displacement Field(VDF). The sampled points are first encoded with PointNet to extract point-wise features, which are then aggregated into voxel-level features through pooling. A sparse Transformer encoder is subsequently applied to produce the latent representation. Next, we employ a sparse Transformer decoder, followed by a pruning operation to predict vertices. The predicted vertices are further refined through subdivision to obtain higher-resolution geometry. Finally, a connection head is introduced to determine the connectivity between pairs of vertices. The overall pipeline is illustrated as follows:

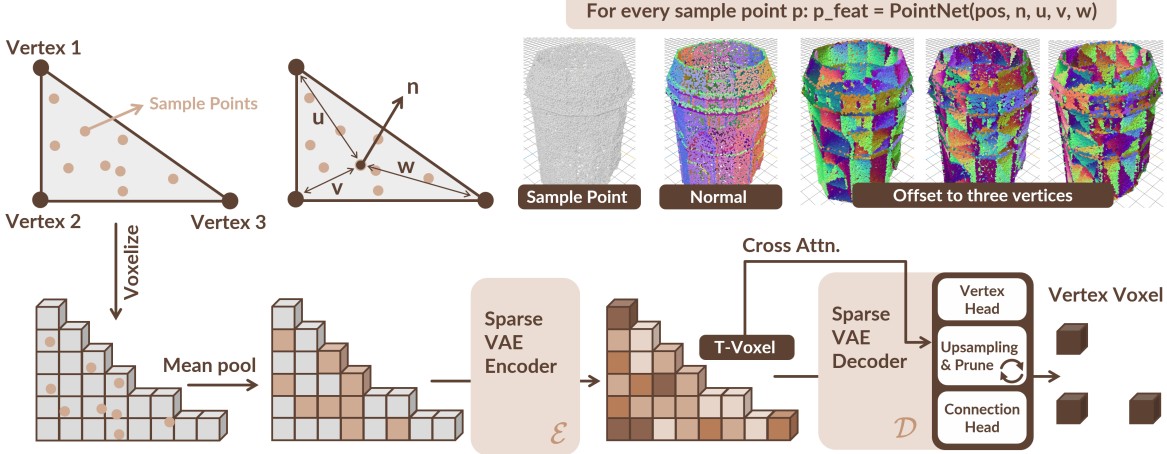

*Figure 11.* Pipeline Details.

## B. Inference Time Comparison Details

We detail the inference time and the generated face in comparison with autoregressive generation models in Tab. 5

*Table 5.* Relationship between inference time and actual output face number for different methods. Each method performs single-batch inference on an H100 GPU across 200 meshes with varying face numbers. Selected key points are displayed here for illustration. The coordinates of each point represent the inference time and the actual output face number, respectively.

| Generation Type | Method | Point-0 | Point-1 | Point-2 | Point-3 | Point-4 |
|---|---|---|---|---|---|---|
| Autoregressive | MeshAnythingv2 | (13.65, 460) | (22.87, 661) | (39.84, 1211) | (50.31, 1612) | (70.86, 2001) |
| | MeshSilkSong | (42.23, 638) | (117.0, 1807) | (180.3, 2896) | (219.2, 3558) | (299.9, 4283) |
| | FastMesh | (5.289, 4001) | (7.636, 6787) | (10.66, 9475) | (16.09, 11426) | (19.02, 14666) |
| | BPT | (36.85, 623) | (81.25, 1404) | (109.4, 1995) | (164.2, 3190) | (251.8, 4029) |
| | DeepMesh | (147.5, 3526) | (241.2, 5355) | (415.8, 9242) | (484, 12084) | (565.7, 13737) |
| Flow-based | **Ours** | (**5.470**, 265) | (**5.031**, 2702) | (**5.298**, 5262) | (**4.857**, 9803) | (**9.873**, 14809) |

## C. More Qualitative Results

Here we provide more qualitative results.

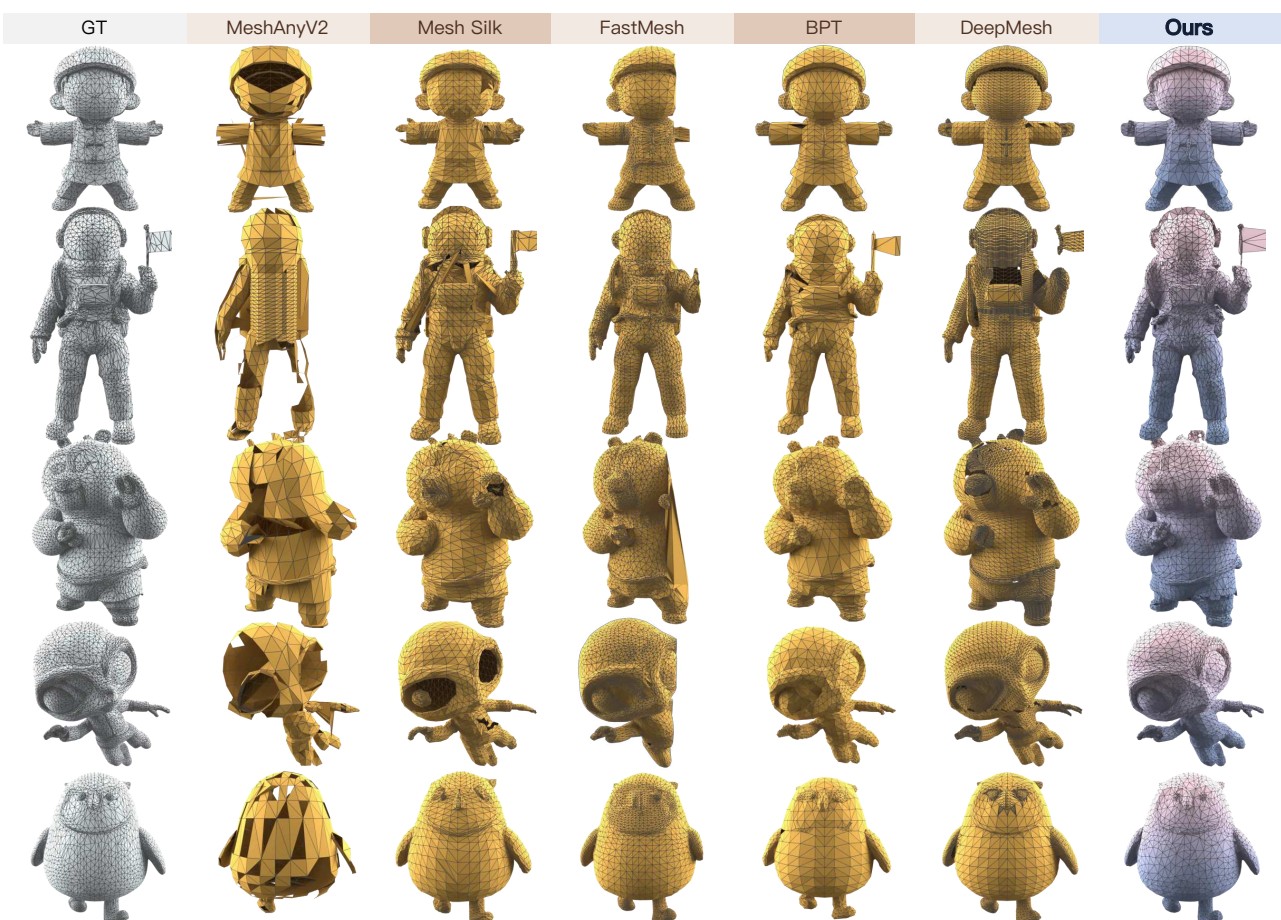

*Figure 12.* More qualitative results on geometry conditioned generation comparison with autoregressive model.

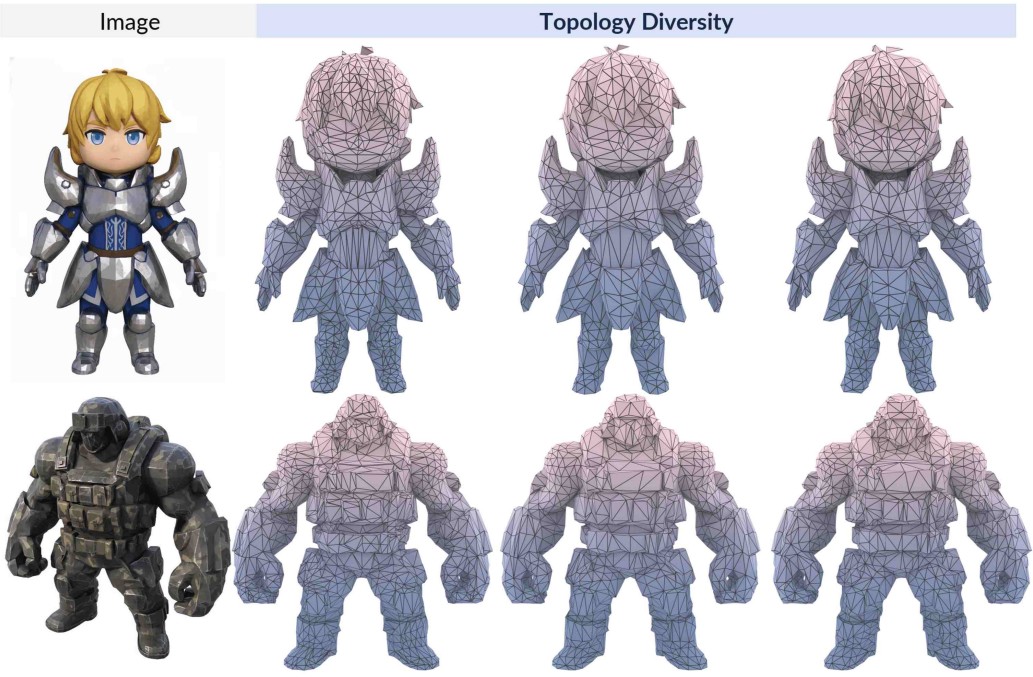

*Figure 13.* Qualitative results on topology diversity.

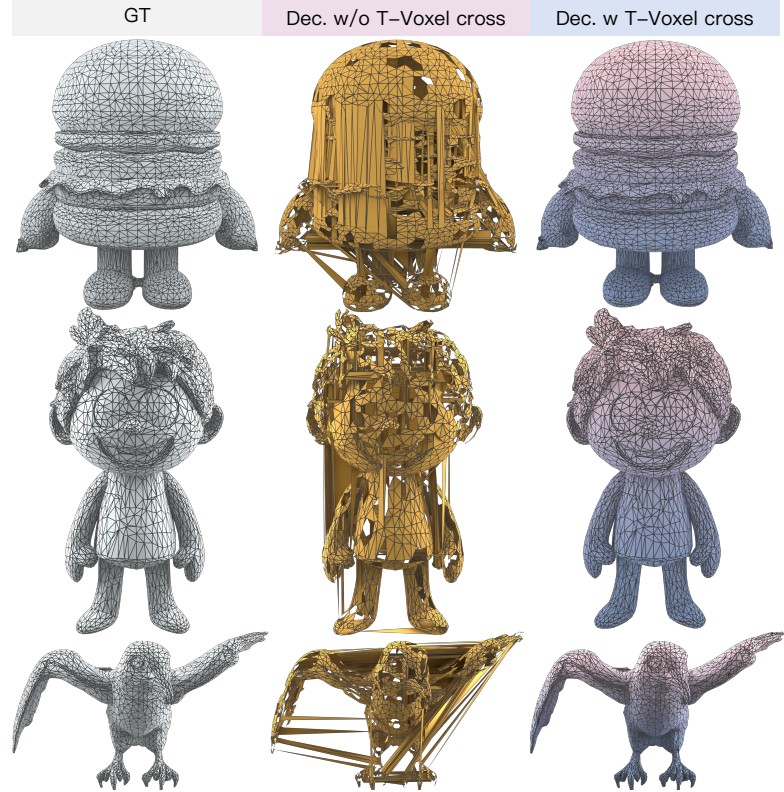

*Figure 14.* More qualitative results of ablation study on T-Voxel cross-attention in decoder.

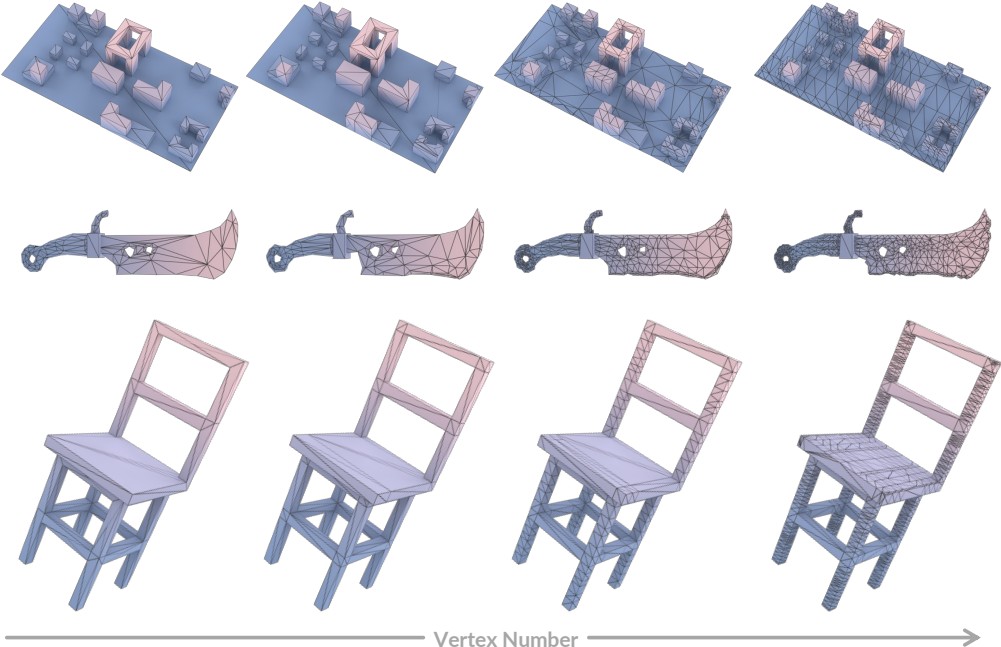

*Figure 15.* More qualitative results on effect of vertex number condition.

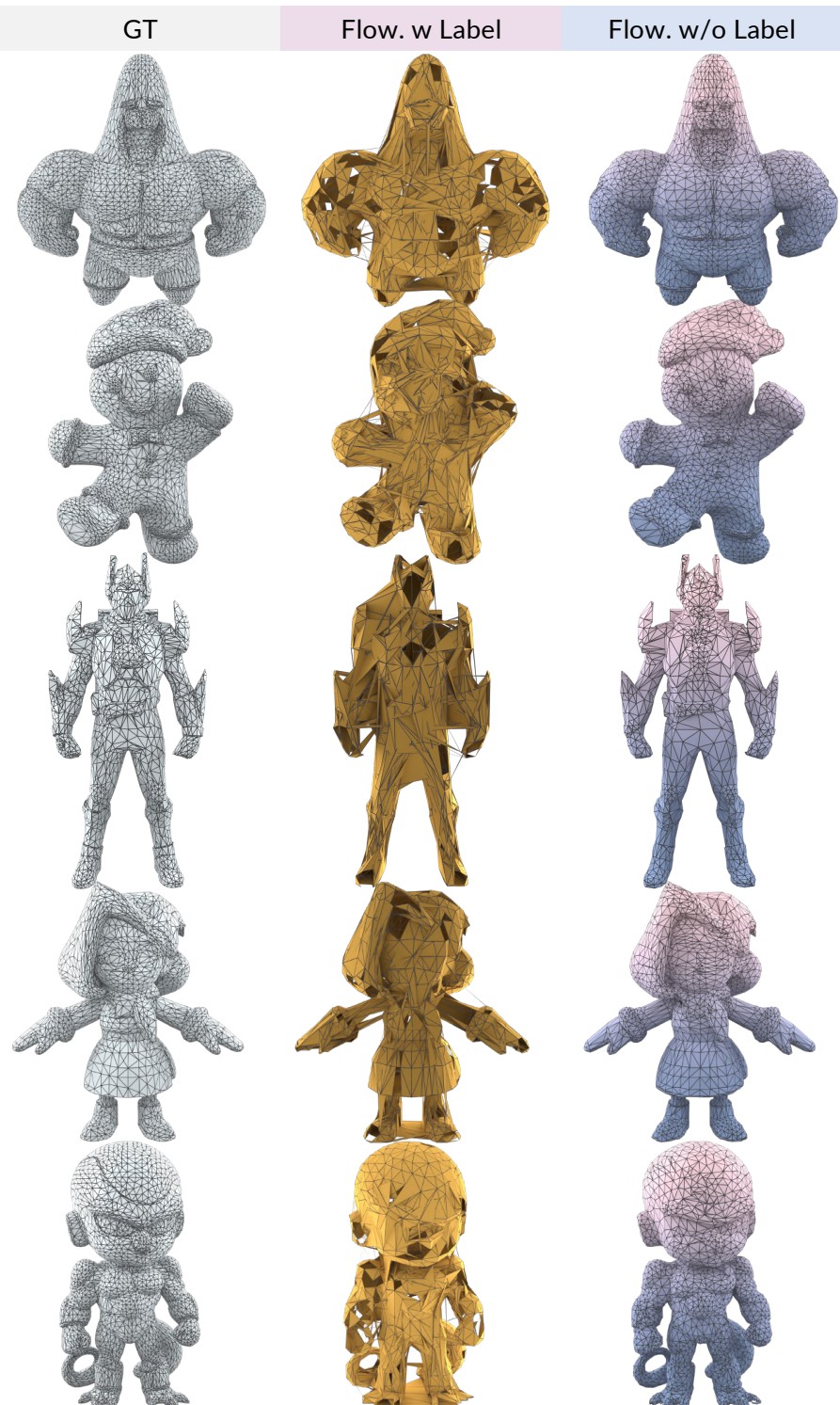

*Figure 16.* More qualitative results of ablation study on VDF.

