# OpenReview forum: "LATO: 3D Mesh Flow Matching with Structured TOpology Preserving LAtents"
_ICML.cc/2026/Conference — ICML 2026 regular_

### Official Review · Reviewer_ktk9 · 2026-03-13

**Soundness:** 3
**Presentation:** 4
**Significance:** 4
**Originality:** 4
**Overall Recommendation:** 5
**Confidence:** 3

**Summary:**

This paper introduces a topology‑preserving strategy that encodes meshes as vertex displacement fields compressed into a structured voxel-based latent. A VAE decoder progressively refines these voxels into precise vertex positions and predicts mesh connectivity to explicitly model mesh topology. Using a two‑stage flow‑matching process, the presented approach generates meshes with complex geometry and clean topology from point clouds or 2D images.

**Compliance With Llm Reviewing Policy:**

Affirmed.

**Final Justification:**

The paper presents a good contribution, and the authors answered my main concerns.

**Key Questions For Authors:**

See weaknesses above.

**Limitations:**

The paper does not seem to pose any potential negative societal impact.

Some limitations are addressed in the conclusions section of the paper.

**Strengths And Weaknesses:**

### Strengths

1. A significant part of the novelty of the presented method relies on the Vertex Displacement Field (VDF). Although simple, the VDF is an interesting approach to defining a continuous vector field anchored to the topology that mathematically highlights the location of the original's mesh vertices, i.e., when the $\mathcal{F} = 0$.
2. The paper is very well written, and it is easy to follow. The proposed method and the notation make it a really easy manuscript to read, with a clear and sound explanation of the relevant concepts.
3. The paper was evaluated in highly complex 3D models beyond synthetic examples in state-of-the-art datasets. The results back the claim that the author's method produces artist-crafted-like explicit meshes with high accuracy comparable to implicit 3D reconstruction methods.

### Weaknesses

1. While the authors state that their method is designed to capture open geometries and non-manifold meshes, the latter properties in the predicted meshes are important to guarantee their usability for rendering, gaming engines, etc. It is not clear (especially from the Loss functions) how the proposed method guarantees that the presented method does not predict self-intersecting meshes or isolated faces. Self-intersections pose important problems for 3D reconstruction (See papers such as Pixel2Mesh [1] and MeshRCNN[2]). The cross attention in the decoder. Figures 10 and 14 seem to be related, but a further explanation is needed.
2. The "Image to 3D Mesh" experiment needs further clarification. For instance, it is not clear how the authors of their "structure flow matching model to synthesize a structure voxel from images." The connection between an input 2D image and the VDF approach to define the T-voxel features is not clear.


[1] Wang, N., Zhang, Y., Li, Z., Fu, Y., Liu, W., & Jiang, Y. (2018). Pixel2Mesh: Generating 3D Mesh Models from Single RGB Images. ArXiv, abs/1804.01654.
[2] Gkioxari, G., Malik, J., & Johnson, J. (2019). Mesh R-CNN. Proceedings of the IEEE/CVF International Conference on Computer Vision (ICCV), 9785–9795.

---

> ### Author Rebuttal · Authors · 2026-03-31
>
> We sincerely thank you for your time and insightful review. We are highly encouraged by your recognition of our VDF representation as an **interesting approach**, writing as **clear and sound**, and **empirical results effectively backing the claim** that LATO generates high-quality, artist-crafted-like meshes. We believe LATO provides a **pioneering, highly efficient paradigm** for direct mesh generation that will serve as a robust baseline for future work. We address your specific concerns below:
>
> ## **W1: Self-intersecting Meshes or Isolated Faces**
> Preventing self-intersections and isolated faces is crucial for downstream usability. Methods such as Pixel2Mesh and Mesh R-CNN mitigate these issues using auxiliary training losses (e.g., Laplacian smoothing to prevent abrupt vertex penetration, or edge-length regularization to keep the geometry compact).
>
> In contrast, LATO prevents self-intersections inherently through its Voxel Distance Field (VDF) representation. Because the VDF is continuous and strictly anchored to the surface, any point on a valid mesh unambiguously points to exactly the three vertices defining its local triangular face. If self-intersection occurs, the spatial region of the intersection would force the VDF to simultaneously point to two conflicting sets of vertices. Since a deterministic network cannot generate such multi-valued, contradictory fields, self-intersecting geometries are fundamentally out-of-distribution for our model.
> Furthermore, our VAE is explicitly supervised to distill this valid VDF back into discrete connectivity, achieving near-perfect F1 scores (99% for vertices, 98% for edges). Consequently, with sufficient training, our topology flow matching model learns a latent space of strictly valid, well-formed geometries, naturally avoiding self-intersections and isolated faces without needing auxiliary losses. For demonstration, we provde visual comparison in Fig. [`R3_W1`](https://anonymous.4open.science/r/lato-66E1/reviewer3/r3_w1.md). As shown, removing the VDF input leads to severe self-intersections, whereas using VDF effectively prevents them.
>
> ## **W2: Clarification on Image to 3D Mesh Experiment**
>
> We apologize for the ambiguity regarding the image to 3D inference pipeline. To clarify, the VDF is strictly a training-time representation used by our VAE to learn the latent T-Voxel space; it is not directly computed from the 2D image during inference.
>
> The complete image-to-3D inference pipeline operates in two stages:
>
> 1. Structural Flow Matching: A single input image conditions our structure flow model to generate a $32^3$ dense latent representation, which is then decoded into a $128^3$ binary occupancy voxel grid.
> 2. Topology Feature Flow Matching: We extract the active voxels (occupancy > 0.5) from the binary occupancy voxel grid, and format them as a sparse tensor. Conditioned on this coarse structural proxy, our topology feature flow model predicts the latent T-Voxel feature for each active voxel. Finally, the VAE decoder transforms these T-Voxels into the explicit topological mesh.
>
> In summary, the image dictates the structural voxels (Stage 1), while the topology flow model synthesizes the fine topological connectivity feature (derived from VDF) (Stage 2). We will explicitly detail this two-stage inference process in the revised manuscript.

---

> > ### Author Rebuttal · Reviewer_ktk9 · 2026-04-01
> >
> > Thank you, the rebuttal clarified most of my questions. I advise including the rebuttal text about the self-intersection in the main text or the supplemental material.
> >
> > I stay firm with my "accept" rating.

---

> > > ### Author Response · Authors · 2026-04-04
> > >
> > > We sincerely thank you for your strong support. Per your valuable advice, we have incorporated the detailed explanation of how our VDF representation inherently prevents self-intersections and isolated faces into the revised manuscript. We have also provided detailed explanations on image to 3D mesh pipeline. Thank you again for your time and constructive suggestions!

---

### Official Review · Reviewer_cTd7 · 2026-03-13

**Soundness:** 2
**Presentation:** 3
**Significance:** 3
**Originality:** 3
**Overall Recommendation:** 4
**Confidence:** 4

**Summary:**

The paper introduces a 3D generative framework designed to synthesize explicit meshes with preserved topology. LATO represents meshes by sampling surface points that encode displacement vectors pointing to their enclosing face vertices. A sparse voxel VAE compresses the VDF into T-Voxels, which encode both spatial distribution and connectivity of mesh vertices.

**Compliance With Llm Reviewing Policy:**

Affirmed.

**Final Justification:**

The rebuttal is very impressive for me to resolve most of my concerns. I think the results and table could be included in the main paper and the supplement.
Thanks for the further response. The results look good for me and encourage the author to discuss these results in detail in the revision.
The full code should be open source. I have raised the score.

**Key Questions For Authors:**

See weaknesses

**Limitations:**

See weaknesses.

**Strengths And Weaknesses:**

#Strengths

- Unlike implicit models that rely on Marching Cubes, LATO generates artist-friendly edge and well-formed topologies directly, making the meshes more suitable for editing.

- The VDF representation allows the model to train on open-surface and non-manifold assets.


#Weaknesses

- While the paper addresses varying vertex counts via hierarchical pruning, the subdivide and prune strategy's reliance on a 0.5 probability threshold might lead to noise or missing vertices in very thin or complex areas.

- The generation pipeline relies on a pre-trained structure model (modified from TRELLIS). Errors in the initial voxel structure synthesis will inevitably propagate to the topological refinement stage. Could you discuss some failure cases? BTW, since the proposed method depends on the explicit voxel representation, which is hard to represent the open surface. Do you have some explanations?

- Could you try some examples for the open surface generation, such as the cloth, or some complex geometry with open surface, some examples you can refer to the paper "NeAT: Learning Neural Implicit Surfaces with Arbitrary Topologies from Multi-view Images".

- What about the inference time of a whole shape?

- Since there are some models can support the well generated wireframe (Hunyuan3D 2.5, which version are you used in the exps), some comparison with the recent advanced work should also be provided ,such as the new Hunyuan3D model and Trillis 2. If the quality of the wireframe can be evaluated, it is more better.

- How does the model ensure that the predicted edges actually form closed, water-tight faces where intended, rather than floating edges or open gaps?

- How does the number of surface points sampled for VDF encoding affect the T-Voxel representation? Is there a minimum sampling density required to ensure complex topological features (like thin structures) are captured?

- Since the connection head predicts edge existence between pairs, does the model include any post-processing to remove edges that don't contribute to a valid manifold, or is the artist-friendly topology purely a result of the learned weights? Could there be a user study to evaluate the performance of the "artist-friendly" topology?

---

> ### Author Rebuttal · Authors · 2026-03-31
>
> We sincerely appreciate your thoughtful feedback. To clarify our scope, our objective is to **introduce and validate a pioneering, highly efficient paradigm** for topology-preserving direct mesh generation using accessible open-source data and compute, rather than competing with commercial models trained on proprietary data and industrial compute (Table 4).
>
> ## **W1: Pruning Threshold vs. Thin/Complex Areas**
>
> We followed the practice in TripoSF to utilize a 0.5 threshold for voxel pruning. This proves to be reliable for our training data distribution, where 80% of meshes contain fewer than 6K triangles. Our VAE achieves an average of 99.84% accuracy (99.72% recall) for vertex and edge predictions during training. On the unseen test set (200 meshes), it maintains 99.69% vertex precision (recall: 99.66%) and 98.24% edge precision (recall: 98.21%). And most errors occur within thin and complex areas.
>
> Hence, **the threshold method is sufficient to validate our explicit mesh flow matching paradigm**. We acknowledge that for extremely thin or complex meshes, noise or missing vertices may occur (as in [`R2_W1`](https://anonymous.4open.science/r/lato-66E1/reviewer2/r2_w1.md)). However, we view LATO as a pioneering basis for future works, and plan to incorporate octree VAE to address more complex structures.
>
> ## **W2: Error Propagation and Open Surface**
>
> Structural errors in initial voxels do propagate. Fig. [`R2_W2`](https://anonymous.4open.science/r/lato-66E1/reviewer2/r2_w2.md) illustrates two failure cases: (A) generates a four-legged chair for a three-legged one, and (B) a double-layered surface collapsing into a single layer due to voxel resolution limits.
>
> Despite, the topology flow model is **highly resilient to localized voxel noise**. To demonstrate, we randomly added and removed 1%, 5%, and 10% of the structural voxels. Fig. [`R2_W2(C)`](https://anonymous.4open.science/r/lato-66E1/reviewer2/r2_w2.md) shows the model consistently decodes valid, clean meshes.
>
> Regarding open surfaces, our $128^3$ T-Voxel is fine enough to distinguish open boundaries from closed volumes, see W3 for generated open surfaces.
>
> ## **W3: Open Surface Generation Results using Cases in NeAT**
>
> **LATO natively generates open surfaces**, as evidenced by the ground (Figs. 8, 9) and flag (Fig. 12) in paper. Per your suggestion, we further validated using two clothing assets from NeAT. Fig. [`R2_W3`](https://anonymous.4open.science/r/lato-66E1/reviewer2/r2_w3.md) demonstrates that LATO effectively reconstructs complex open boundaries.
>
> ## **W4: Inference Time of a Whole Shape**
>
> On a NVIDIA H100, generating a shape from scratch takes 21 to 29 secs. The breakdown is: structure flow model ~19 secs; topology feature flow 2-7 secs (scaling with active voxels); and decoding T-Voxels into a mesh 0.2-3 seconds.
>
> ## **W5: Comparison with New Hunyuan Model and Trellis 2**
>
> We evaluated on Hunyuan3D 3.0 (through web). Per your suggestion, we expand to Hunyuan3D 3.1, Hunyuan3D lowpoly-V1.5, and TRELLIS.2. Fig. [`R2_W5`](https://anonymous.4open.science/r/lato-66E1/reviewer2/r2_w5.md) provides a visual comparison. For quantitative comparison, we adopt Topology Score in MeshRFT (R1W1). LATO outperforms implicit meshes and is comparable with lowpoly-V1.5, despite model size and training resources disparities.
>
> **Table 6**: Comparison of Topology Score
> |Methods|Topology Score↑|
> |:-|:-:|
> |Hunyuan3D-V3.1|0.3066|
> |Hunyuan3D lowpoly-V1.5|**0.5561**|
> |TRELLIS.2|0.5086|
> |LATO|0.5551|
>
> ## **W6: Floating Edges or Open Gaps**
>
> During decoding, we form faces by extracting triangular edge loops (i.e., v1-v2-v3-v1). Any edge that does not contribute to a triangle is discarded, and no further post-processing. As our model achieves high edge prediction accuracy (as detailed in W1), floating/gap edges are very rare. We attribute this structural integrity to our surface-anchored VDF representation. The network is explicitly supervised to distill vertex and edge connectivity directly from the VDF, where floating edges would unnaturally deviate from the surface, and open gaps would inherently contradict the continuous VDF.
>
> ## **W7: Effect of Sampled Surface Points**
> The below details ablation on sampling density.
>
> **Table 7**: Effect of Sampled Surface Points
> |Num of Points|Vert Precision↑|Vert Recall↑|Edge Precision↑|Edge Recall↑|
> |:-:|:-:|:-:|:-: |:-:|
> |81920|0.9392|0.9346|0.9147|0.9133|
> |204800|0.9745|0.9738|0.9589|0.9581|
> |409600|0.9884|0.9879|0.9741|0.9750|
> |819200|**0.9954**|**0.9956**|**0.9817**|**0.9822**|
>
> In addition, to capture thin structures, we conduct hybrid sampling: 400K points sampled uniformly across the surface, and 400K distributed evenly within each individual face, see [`R2_W7`](https://anonymous.4open.science/r/lato-66E1/reviewer2/r2_w7.md) for ablation on this.
>
> ## **W8: Non-contributing Edges and User Study**
>
> Please see W6 for edge processing, see **Table 2** (in reponse to W1 R1) for the user study questionnaire and results.

---

> > ### Author Rebuttal · Reviewer_cTd7 · 2026-04-01
> >
> > Thanks for the rebuttal. The rebuttal is very impressive for me to resolve most of my concerns. I think the results and table could be included in the main paper and the supplement. There is still an issue on open surfaces: I notice that the results on the open surface are not regular compared to the GT; I want to see the box result of the teaser of NEAT. And for the open surface result in Fig. R2_W2, I think the generated result is good (it is truly an open surface compared to GT); GT is the double surface for the cloth. For these cases, I still want to see some result on an open surface with more complex shapes.
> >
> > Based on this, I am willing to raise the score to positive.

---

> > > ### Author Response · Authors · 2026-04-04
> > >
> > > We are deeply encouraged by your positive feedback and sincerely appreciate your willingness to raise the score. Per your suggestion, we have officially **integrated the Topology Score table (Table 1), the User Study results (Table 2), and a more explicit discussion of open surface generation** into the main paper. We have also **cited NeAT and included comparisons using its open surface cases**. We sincerely thank you for these constructive suggestions, which have strengthened our submission.
> > >
> > > Regarding the boundary irregularity of open surfaces, we conducted a deeper analysis. We observed that **boundary regularity varies across different generated samples**: for the exact same garment case, some generated samples exhibit highly regular boundaries while others appear less smooth, see Fig. [`garment_samples`](https://anonymous.4open.science/r/lato_rebuttal-54B1/reviewer2/Diversity.md). Moreover, LATO consistently generates regular boundaries for certain categories (e.g., the ground in Figs. 8 and 9, and the flag in Fig. 12 of the main paper). Therefore, we conclude that this issue is not a fundamental limitation, but rather **an artifact caused by the training data distribution**.
> > >
> > > Specifically, open surfaces, and garments in particular, comprise only a small fraction of our training set. When generating these underrepresented geometries, the vertices decoded from our $128^3$ voxel latent into the high-resolution $1024^3$ space can **occasionally deviate from perfect boundary alignment due to insufficient structural learning** from the training data.
> > >
> > > To validate this and also follow your suggestion, we evaluated **LATO on the Box and Catmask cases from the NeAT teaser**, as shown in Fig. [`box+catmask`](https://anonymous.4open.science/r/lato_rebuttal-54B1/reviewer2/NEAT_Teaser.md). For the Box, LATO generates highly regular, straight boundaries, benefiting from abundant similar geometric priors in the training data. For the Catmask, LATO successfully captures the open surface structure, with boundary smoothness performing well within the expected quantization limits.
> > >
> > > We further evaluated **LATO on a variety of highly complex open surface models**, as in Fig. [`complex_opensurface`](https://anonymous.4open.science/r/lato_rebuttal-54B1/reviewer2/Complex.md). For each case, we provide two diverse generated samples to highlight our model's ability to synthesize varying topologies. In cases (A), (B), and (C), LATO successfully generates high-quality geometry with well-preserved, clean open boundaries. Case (D) features an extremely complex open surface (a detailed plant). Remarkably, LATO still successfully generates the open surfaces for the individual leaves. While some exceedingly thin tree branches are omitted (which is consistent with the resolution bottleneck discussed in our previous response to W1), the overall open-surface topology is well preserved.
> > >
> > > We hope these additional results fully resolve your remaining questions, and we thank you again for your time and constructive feedback.

---

### Official Review · Reviewer_jbkB · 2026-03-16

**Soundness:** 2
**Presentation:** 3
**Significance:** 2
**Originality:** 3
**Overall Recommendation:** 4
**Confidence:** 4

**Summary:**

The paper studies explicit 3D mesh generation, aiming to generate meshes with both detailed geometry and well-formed topology. Its core contribution is LATO, a topology-preserving sparse voxel latent representation built from a Vertex Displacement Field (VDF), which encodes for each sampled surface point the displacements to the vertices of its enclosing face. It proposes to adapt a TRELLIS-style backbone for mesh generation. Experimentally, LATO reports stronger reconstruction and generation quality than prior explicit mesh baselines, while also being much more efficient at inference than autoregressive methods and producing more artist-friendly topology than implicit methods.

**Compliance With Llm Reviewing Policy:**

Affirmed.

**Final Justification:**

Good rebuttal, score adjusted.

**Key Questions For Authors:**

More direct quantitative comparison against TRELLIS would help clarify the main tradeoff relative to the most relevant neighboring paradigm: is the gain primarily in topology quality, or also in geometry fidelity and efficiency?

More direct topology-aware metrics could be included to better support the central claim.

**Limitations:**

While the sampling-based training strategy is preferable to truncating mesh sequences, it would be helpful to quantify how closely it approximates full pairwise supervision (quality vs. sampling ratio).

Additional failure-case analysis, beyond the currently acknowledged low-resolution limitation, would help clarify where the method still breaks down in topology prediction or geometric detail recovery.

Typos:

- Sec 4.2: matchign: matching
- Fig 13: tolpology → topology

**Strengths And Weaknesses:**

Strengths:

1. Intuitive representation. VDF gives a topology-aware dense signal.
2. Faster inference than autoregressive baselines thanks to a TRELLIS-based framework.

Weaknesses:

1. While the use of artistic meshes as targets is appropriate and makes the benchmark more relevant, the current quantitative metrics still mainly capture surface alignment rather than explicit topology quality. Since meshes with very different connectivity can achieve similar CD/HD/NC values, more direct topology-aware or mesh-usability metrics would be needed to fully support the paper’s central claim.
2. Quantitative comparison against strong neighboring implicit baselines such as TRELLIS is still missing.
3. Resolution bottleneck. As acknowledged by the authors, the current representation is still limited by the underlying sparse voxel resolution. In particular, it appears less capable of preserving very fine-grained small parts than some competing models, even when the overall topology is cleaner (Fig. 5).
4. Some comparisons, especially against large implicit foundation models, are mostly qualitative, so the practical tradeoff between topology quality and overall geometry quality is not fully quantified.

---

> ### Author Rebuttal · Authors · 2026-03-31
>
> We sincerely appreciate your time and constructive feedback, along with your recognition of LATO's **intuitiveness** and **computational efficiency**. Before addressing your specific concerns, we wish to clarify that the objective of our paper is to introduce **a pioneering, highly efficient paradigm** for topology-preserving direct mesh generation, and to validate its core effectiveness using accessible open-source data and compute resources, rather than matching the fine geometric details and generalizability of large-scale models trained with industrial-level compute and proprietary data (as in Table 4).
>
> ## **W1 & W4 & Q2: Topology-aware Metrics**
> Thanks for the valuable suggestion. Quantitative evaluation of topology remains an open problem. Most SOTA methods (MeshGPT, MeshAnything V1/V2, DeepMesh, etc.) rely on user study to gather human subjective evaluation. The only exception is MeshRFT, which proposed a metric called **Topology Score** by converting mesh into quadrilateral and evaluate how well-proportioned the quad faces are. Following your suggestion, we evaluate on both Topology Score and user study against baselines on test data as in the main paper.
>
> **Table 1**: Comparison of Topology Scores
> |Paradigm|Methods|Topology Score↑|
> |-|:-:|:-:|
> |Implicit|TRELLIS|0.5264|
> ||Hunyuan3D|0.2946|
> ||Tripo3D|0.5146 |
> ||CLAY|0.5100|
> | Auto-regressive|DeepMesh|0.5500|
> ||FastMesh|0.5470|
> ||BPT|0.5415|
> ||Mesh Silksong|0.5223|
> |T-Voxel| **LATO**|**0.5551**|
>
> LATO outperforms all baselines in terms of Topology Scores.
> For user study, we constructed a questionnaire containing 20 cases, and asked participants to choose from the generated meshes for each case with the best balanced topology and geometry. (Questionnaire Screenshot in Fig. [`R1_W1`](https://anonymous.4open.science/r/lato-66E1/reviewer1/r1_w1.md)).
>
> **Table 2**: User Study Preference Ratio
> |Paradigm|Methods| User Study Ratio↑ |
> |-| :-: | :-: |
> | Implicit| Hunyuan3D|0.0486|
> ||TRELLIS|0.0447|
> ||Tripo3D|0.1284|
> ||CLAY|0.0311|
> |Auto-regressive|DeepMesh|0.0350|
> ||BPT|0.1751|
> || Mesh Silksong |0.0506|
> |T-Voxel|**LATO**|**0.4864**|
>
> As in Table 2, LATO achieves the highest user preference ratio by a significant margin. Notably, the subsequent two highest scores belong to Tripo3D and BPT, which represent distinct user preferences leaning toward geometric fidelity and topological regularity, respectively.
>
> ## **W2 & W4 & Q1: Quantitative Comparison against Strong Implicit Baselines**
> Table 1 shows topology-aware quantitative metric. Following suggestion, we further compare on geometric metrics, CD/HD scores, with TRELLIS on our test data.
>
> **Table 3**: Comparison of CD and HD Values with TRELLIS
> |Methods|CD(L2)↓|CD(L1)↓|HD↓|
> |-|-|-|-|
> |TRELLIS|0.0453|0.0693|0.0845|
> |**LATO**|**0.0440**|**0.0659**|**0.0818**|
>
> LATO's advantages extend to geometric fidelity as well. We attribute this to our architectural design: LATO utilizes a high-resolution voxel grid $128^3$ for latent representation, whereas TRELLIS compresses into $64^3$.
>
> ## **W3: Resolution Bottleneck and Geometric Details**
> For Fig. 5 in the main paper, baselines use roughly 10x more parameters and massive proprietary datasets (Table 4). In contrast, LATO uses only 180M parameters and open-source data (mostly <6K triangles) collected from Objaverse. Consequently, LATO prioritizes low-poly topology over high-frequency geometric detail. Our goal is to demonstrate a fundamentally novel capability in explicit topology generation, instead of competing on sheer geometric fidelity.
>
> **Table 4**: Comparison of Params, Compute with Models in Fig. 5 (* means commercial model)
>
> |-|TRELLIS|Hunyuan 3D*|Tripo3D*|CLAY*|LATO|
> |:---|:-|:-|:-|:-|:-|
> |Params|2B|>3.3B|~1.5B| >1.5B |180M|
> |GPU Time|64xA100 (~10 days)| \ |>64xA100 (~14 days)|>256xA800 (15 days)|8xH100 (7 days)|
>
> Despite resource constraints, results in main paper and rebuttal (Tables 1, 2, 3) demonstrate LATO generates clean, high-quality meshes. Our future work will address fine geometric details by scaling model parameters, incorporating octree VAE, and using denser meshes for training.
>
> ## **L1: Sampling Ratio vs Quality**
> Full pairwise supervision simply leads to out-of-memory. Table 5 provides ablation on number of sampled pairs vs. quality:
>
> **Table 5**: Geometry Quality w.r.t. Sampling Ratio
> | Sampled Pairs (Nn+Nr) | CD(L2)↓ | HD↓ | \|NC\|↑ |
> |-|:-|:-|:-|
> | 48(32+16)| 0.0502 | 0.1115 | 0.7902 |
> | 96(64+32) | 0.0437 | 0.0940 | 0.8337 |
> | 192(128+64) | **0.0402** | **0.0786** | **0.8425** |
> | 384(256+128) | OOM | OOM | OOM |
>
> Performance improves with increased sampled pairs and saturates near 192 points, so we choose 192 to balance memory and performance.
>
> ## **L2: Failure Cases**
> We provide exemplary failure cases in Fig. [`R1_L2`](https://anonymous.4open.science/r/lato-66E1/reviewer1/r1_L2.md), where double layered surface is decoded into a single layer due to voxel resolution, and structural errors in structural flow model propagate.

---

> > ### Author Rebuttal · Reviewer_jbkB · 2026-04-04
> >
> > New experiments provide stronger support for the paper. Will raise the score to weak accept.

---

> > > ### Author Response · Authors · 2026-04-04
> > >
> > > We sincerely thank you for your constructive feedback and for raising the score. We have officially incorporated the new experiments you suggested, including the topology-aware metrics (Table 1), the user study (Table 2), the quantitative comparisons against TRELLIS (Table 3), and the failure cases, to the revised manuscript. Thank you again for helping us strengthen our paper.

---

### Decision · Program_Chairs · 2026-04-30

**Decision:**

Accept (regular)

**Comment:**

This paper presents a topology-preserving framework for explicit 3D mesh generation based on a structured latent representation and flow matching. Reviewers acknowledged the novelty of the representation, the strong technical quality of the paper, and the practical value of directly generating artist-friendly meshes with improved efficiency over autoregressive alternatives. While concerns were raised regarding the evaluation of topology quality / resolution limits / settings details, the rebuttal addressed most of these issues, and all reviewers ultimately converged to positive ratings. Thus, the AC recommends acceptance. The authors should incorporate the clarifications from the rebuttal into the final version.